# MixLLM: Mixed-precision LLM Quantization with Algorithm-system Co-design

## Abstract

Quantization has become one of the most effective methodologies to compress LLMs into smaller size. However, the existing quantization solutions still show limitations of either non-negligible accuracy drop or system inefficiency. In this paper, we make a comprehensive analysis of the general quantization principles on their effect to the triangle of accuracy, memory consumption and system efficiency. We propose MixLLM that explores the new optimization space of mixed-precision quantization between output features based on the insight that different output features matter differently in the model. MixLLM identifies the output features with high salience in the global view rather than within each single layer, effectively assigning the larger bit-width to output features that need it most to achieve good accuracy with low memory consumption. We present the sweet spot of quantization configuration of algorithm-system co-design that lead to high accuracy and system efficiency. To address the system challenge of this sweet spot, we design the two-step dequantization to make use of the int8 Tensor Core easily and fast data type conversion to reduce dequantization overhead significantly. Extensive experiments show that MixLLM achieves the best accuracy on a variety of tasks for the popular LLMs than a set of state-of-the-art works. It shows 0.31 lower perplexity and 0.43% improvement on zero shot tasks for Llama 3 8B than QoQ, with similar memory consumption and system efficiency.

## 1 Introduction

Large language models (LLMs) (Bubeck et al., 2023; Meta, Cited Sep. 2024) have shown remarkable performance on various tasks. But their large memory consumption and massive computation cost have become an obstacle for the efficient deployment (Xia et al., 2023; 2024). Quantization has become one of the most sufficient solution to compress LLMs into smaller size (Frantar et al., 2022; Lin et al., 2024a; Xiao et al., 2023; Yao et al., 2022), by representing the weight or activation with smaller bit-width. However, the existing quantization solutions still show limitations of either non-negligible accuracy drop or system inefficiency.

There is a triangle of characteristics for efficient LLM quantization: *accuracy*, *memory consumption* of parameters, and *system efficiency* of execution, which we call **effectiveness triangle** of quantization. The existing quantization solutions have different focus and trade-off in the triangle:

- The weight-only methodologies target to solve the memory consumption problem, and can speedup the small-batched decoding execution that faces the memory-wall problem (Xia et al., 2023; Kim et al., 2024). But their accuracy drop of 4-bit quantization can be a challenge for the production workloads sensitive to accuracy, as illustrated in recent studies (Wu et al., 2023). Besides, the weight-only method can lead to system performance drop for large-batched workloads (e.g., the SOTA W4A16 kernel only achieves 83% performance of its float16 counterpart at batch size 512 with hidden size 4096, shown in Fig.2).

- The weight-activation quantization represents the activation with low-bit values along with the weights, potentially lead to higher system efficiency. But it can lead to larger accuracy drop than the weight-only method as the activation is usually harder to quantize (Zhao et al., 2024; Ashkboos et al., 2024; Lin et al., 2024b). Besides, it introduces more dequantization overhead for the activation that can hurt the system efficiency. The transformation optimizations in some works can make the system efficiency even worse.

- Outlier separation and mixed-precision technologies emerge to improve the accuracy of low-bit quantization by either excluding the unstructured high-salience weights from quantization (Dettmers et al., 2024; Kim et al., 2024) or assigning larger bit-width for the quantization of structured high-salience weights (Zhao et al., 2024). The former shows system efficiency problem due to the low efficiency of half precision (i.e., float16/bfloat16) sparse tensor processing. The state-of-the-art mixed-precision solution (Zhao et al., 2024) aims for low-bit quantization but shows non-negligible accuracy drop, even inferior to the 4-bit weight-only quantization.

**Contributions.** In this paper, we provide an extensive analysis of the general quantization principles. To address the limitations of the previous works and cover the three characteristics in the effectiveness triangle, we propose MixLLM, which makes the following contributions:

▶ **High accuracy with low memory consumption: mixed-precision between output features on the weight, with global salience identification.** Given that different neurons matter differently to the model's output, we use different bit-width for different output features (i.e., output channels) for the weight quantization, 8-bit for output features with high salience and 4-bit for others. Rather than using a uniformed number of outliers within each layer according to the estimated salience w.r.t. each single layer (Zhao et al., 2024), MixLLM identifies the salience of different output features globally according to the estimated loss to the model's output. This is because different layers can have different importance to the model. Besides, the mixed-precision between output features makes the system design easier than between input features because the calculation of different output features are disjoint sub-problems.

▶ **High accuracy with good system efficiency: the co-designed quantization configuration and GPU kernel optimization.** We observe the sweet spot of several quantization decisions to achieve both good accuracy and system efficiency. MixLLM uses 8-bit for activation quantization as it can retain a good accuracy. Besides, MatMul execution tends to be bound more on the larger weight tensor rather than the smaller activation tensor, which weakens the need to push the activation smaller (refer to Sec.3.1). MixLLM uses symmetric quantization for 8-bit and asymmetric for 4-bit for good accuracy, both in group-wise manner. Such configuration makes it challenging to achieve good system efficiency. We design the two-step dequantization to enable using fast int8 Tensor Core for such configuration, along with the fast integer-float conversion to decrease the dequantization overhead.

## 2 BACKGROUND, RELATED WORK, AND DISCUSSION

### 2.1 BACKGROUND OF QUANTIZATION

The quantizaiton maps the tensor $X$ into the target range with smaller bit-width representation through affine transformation: $X_q = clamp(\lfloor \frac{X}{s} \rceil + z, range)$, where $s$ is the scale and $z$ is the zero point. The value can be recovered (i.e., dequantization) through: $X' = (X_q - z) \times s$. $X'$ is pushed to the discrete chunks rather than recovered to the original value, thus has accuracy loss. The bit-width is essential for the accuracy of quantization as it determines the number of chunks for the quantized values ($2^{bit\_width}$). Take an example, enlarging the bit-width from 4 to 5 can double the number of chunks, so that the 5-bit RTN quantization can easily beat the 4-bit quantizations with advanced techniques (Tab.1).

The scale and zero point can be calculated from the whole channel/token vector or a small group within the channel/token, the former is called per-channel/token quantization and the latter is group-wise quantization. The group-wise scheme results in smaller accuracy loss due to the smaller chunk scale, but requires more complex GPU kernel design[1].

The symmetric quantization uses 0 as the zero point value, which simplifies the computations ($X_q = clamp(\lfloor \frac{W}{s} \rceil, range)$, $X' = X_q \times s$). This simplification enables many works to design the per-channel/per-token quantized linear kernels by multiplying the scales at the epilogue of the whole MatMul (matrix multiplication) for dequantization (Xiao et al., 2023; TensorRT-LLM, Cited Sep. 2024). However, the symmetric quantization usually leads to larger loss than the asymmetric one as the data distribution can be usually asymmetric, especially for smaller bit-width like 4-bit.

---

[1]We mainly discuss the model execution on the GPU in this paper. But the basic principle is general.

## 2.2 RELATED WORKS AND DISCUSSION OF GENERAL QUANTIZATION PRINCIPLES

This paper mainly focuses on post-training quantization (PTQ).

**Systems that affect the quantization requirement.** The continuous batching technology (Yu et al., 2022) enables to batch the decoding tasks from different requests together to enlarge the batch dimension of MatMul during LLM inference. The SplitFuse method (Holmes et al., 2024) advances the continuous batching by merging the prefill and decoding tasks into the same batch, further enlarging the MatMul shapes. These technologies pushes the server side LLM jobs to become the compute-bound workloads and further motivate the demand to reduce the massive computation.

**Weight-only quantization and its limitation.** There emerges a wide range of technologies to improve the accuracy of weight-only quantization. GPTQ (Frantar et al., 2022) advances OBC (Frantar & Alistarh, 2022) on OBS-based (Hassibi et al., 1993) weight compensation with blocked updating and reordering. AWQ (Lin et al., 2024a) proposes to scale the weight according to the characteristic of activation. OminiQuant (Shao et al., 2024)) proposes the learnable scaling and weight clipping factors. SpQR (Dettmers et al., 2024), SqueezeLLM (Kim et al., 2024) and OWQ (Lee et al., 2024) separate the outliers from the quantiation and with half precision. QuiP (Chee et al., 2023) aims to achieve extreme low-bit quantization with incoherence processing. ZeroQuant(4+2) (Wu et al., 2023) aims to improve accuracy with medium-sized FP6 quantization.

The weight-only quantization does not reduce the computation but introduces the extra dequantization operations. The low-bit weight will be dequantized to float16 to execute the MatMul in float16 datatype. The current weight-only quantization faces two challenges: 1) From the accuracy aspect, there is still an accuracy gap between the 4-bit quantization and the float16 model, especially for many real business scenarios sensitive to the small accuracy drop, as discussed in the recent works (Wu et al., 2023; Xia et al., 2024). 2) It can lead to system efficiency drop on busy servers as the recent LLM inference serving systems will usually batch the processing of different requests together on the server and form large MatMuls. The large MatMuls are compute-bound and will suffer from the dequantization overhead (Lin et al., 2024b).

**Weight-activation quantization and the challenges.** The weight-activation quantization helps to make use of the low-bit computing unit. LLM.int8() (Dettmers et al., 2022) observes the activation outlier problem and separates outliers from quantization with half precision. ZeroQuant (Yao et al., 2022) proposes the per-token activation quantization and group-wise weight quantization. SmoothQuant (Xiao et al., 2023) addresses the activation outlier problem through smoothing, and AffineQuant (Ma et al., 2024) proposes the general affine transformation for quantization. RPTQ (Yuan et al., 2023) reorders the channels to cluster similar scaled values together. SpinQuant (Liu et al., 2024) and QuaRot (Ashkboos et al., 2024) leverages matrix rotation properties to alleviate the outlier phenomenon. Atom (Zhao et al., 2024) uses the mixed-precision between input features to improve accuracy of 4-bit activation quantization. QoQ (Lin et al., 2024b) is a holistic weight-activation-KV quantization solution with progressive group quantization, attention smoothing, and channel reordering.

Even though the weight-activation quantization has the advantage of reduced MatMul computation (i.e., MatMul in smaller bit-width to make use of the smaller bit-width computing unit with higher computation throughput[2]), it faces the challenge of accuracy drop caused by the activation quantization, especially that the activation is usually harder to quantize than the weight. The SOTA low-bit weight 8-bit activation solution (W4A8) (Lin et al., 2024b) still have a gap to the 4-bit weight only quantization. Beside the accuracy drop, the activation quantization will introduce more dequantization overhead than the weight-only one, which is another challenge from system side.

The existing solutions focus on partial of the effectiveness triangle, but cannot cover all of them well. MixLLM is orthogonal to the above works by exploring the mixed-precision between output features with global salience identification, and the co-designed quantization decision and GPU kernels.

---

[2]The extra dynamic activation quantization kernel can be fused into other operators with very little system cost (Zhao et al., 2024), thus we only discuss the MatMul itself.

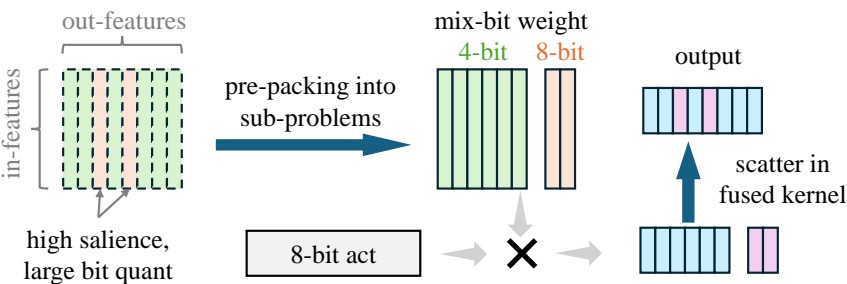

Figure 1: Illustration of mixed-precision between output features.

## 3 METHODOLOGY

### 3.1 QUANTIZATION DESIGN AND DECISION IN MIXLLM

To cover the three aspects of the effectiveness triangle simultaneously, we make the following design and decision of weight and activation quantization according to the analysis in Sec.2.2.

**A. Mixed-precision between output features of weight, with global salience identification.**

It is known that different elements of the weight show different salience to the network's loss when being quantized (Kim et al., 2024; Dettmers et al., 2024). The outlier separation method can improve the accuracy by using float16 to store the high-salience elements, but suffers from the system efficiency problem (detailed analysis in Sec.A.4). We observe that the elements with high salience tend to show distribution along the output channels for most of the linear layers in many LLMs. Based on this observation, we can assign larger bit-width to the output channels of high salience, and smaller bit-with to the others, forming structured mixed-precision quantization. Through the experiments, we get the same conclusion with the existing works (Kim et al., 2024; Dettmers et al., 2024) that there is only a small set of elements with high salience contributing significantly to the model's accuracy drop. Thus we only need to assign the large bit-width to a small portion of the output channels to achieve good accuracy and retain a small memory consumption at the same time.

The structured mixed-precision between different output channels can be friendly to the system efficiency and kernel development, due to the nature that different output features are disjoint in the MatMul and the computation of them are different sub-problems. Fig.1 shows how the linear layer computes with the mixed-precision between output features. It divides the linear into independent sub-problems, and finally gathers the output of the sub-problems together to form the final result. This optimization space is orthogonal to the existing quantization optimizations, e.g., GPTQ (Frantar et al., 2022), and can be applied together with them.

One critical problem is how to identify the high-salience output channels in the model. The fixed threshold (Dettmers et al., 2024) or the fixed number/ratio (Zhao et al., 2024; Lee et al., 2024) of high salience elements computed by the local loss of layers can be sub-optimal to the end-to-end model, as different layers can show different importance to the model's final output (Gromov et al., 2024; Men et al., 2024; Dong et al., 2019). A high salience channel w.r.t. a layer may not be a high salience channel of the end-to-end model. In MixLLM, we compute the high salience channels globally according to their impact to the model's final loss (Sec.3.2). As a result, different layers will have different number of high salience channels.

Note that this design is different from the mixed-precision in Atom (Zhao et al., 2024) from two aspects. 1) MixLLM first addresses the problem of identifying the high-salience channels globally rather than locally. 2) MixLLM applies the mixed-precision between output features rather than input features, which is more system performant and algorithm flexible[3] as the output features are disjoint naturally.

**B. Sweet spot of quantization decision with algorithm-system consideration: 8-bit symmetric activation and 4-bit asymmetric weight quantization in group-wise manner.**

---

[3]One example is that, the outlier number in each MatMul should be a multiplier of the corresponding tiling size of kernel design to achieve a good system efficiency in Atom, which limits the flexibility of algorithm.

MixLLM makes the same decision with QoQ (Lin et al., 2024b) on activation quantization to use 8-bit, as the 4-bit activation can lead to a large accuracy drop but does not lead to significant system efficiency improvement as MatMul execution tends to be bound more on the larger weight tensor rather than the smaller activation tensor. It can be partially indicated from the compute intensity of the linear layer. Given token number $M$ and input and output features $K$ and $N$, the compute intensity $I = \frac{2MNK}{MKB_{act}+KNB_{weight}}$. $B_{act}$ and $B_{weight}$ are the bytes per element of activation and weight. Given $M = 512$ and $N = K = 4096$, reducing $B_{weight}$ from 8 to 4 will results in a 80% increasement of $I$, while reducing $B_{act}$ from 8 to 4 will only achieve 5.88% increasement.

Instead of using per-token and smoothing on the activation quantization, MixLLM uses group-wise RTN method. On the one hand, Tab.1 shows that simple group-wise RTN quantization performs better than token-wise smoothing method. On the other hand, the weight is already group-wise in MixLLM, and the group-wise activation does not lead to significant more dequantization overhead in the system. We observe symmetric quantization is enough for the 8-bit activation (refer to MixLLM W8A8 in Tab.1), while asymmetric can be essential for the 4-bit weight. The group-wise method with asymmetric can lead to a difficulty for the kernel to make use int8 Tensor Core, for which QoQ (Lin et al., 2024b) introduces the two-step quantization method. Instead, we design a two-step dequantization with the property of the mix of symmetric and asymmetric (Sec.3.3).

## 3.2 GLOBAL PRECISION SEARCH ALGORITHM

As discussed in Sec.3.1, MixLLM determines the precision of all output features in all layers globally rather than locally. It identifies the salience of these features with respect to the final loss of the model, and assigns larger bit-width to the features leading to larger loss.

Specifically, it calculates the salience $S$ of a channel $c$ as:

$$S_c = |l(c_q) - l(c_0)| \tag{1}$$

which is the distance of the model's loss between quantizing and not quantizing this single channel. In Eq.1, $l()$ is the loss function of the model w.r.t. a single channel, $c_q$ is the quantized weight of the channel and $c_0$ is the original weight. Note that it regards other neurons except $c$ as constant in $l()$.

We use the Taylor Expansion method to estimate the loss function $l(c)$ (similar with the existing quantization works, ignoring the high-order items):

$$l(c) \approx l(c_0) + g^T(c - c_0) + \frac{1}{2}(c - c_0)^T H(c - c_0) \tag{2}$$

where $g = \mathbb{E}[\frac{\partial}{\partial c}l(c)]$ is the loss's gradient w.r.t. the channel, and $H = \mathbb{E}[\frac{\partial^2}{\partial c^2}l(c)]$ is the second-order gradient (i.e., Hessian matrix) w.r.t. the channel.

It is infeasible to calculate the Hessian matrix as it is too costly. We approximate the Hessian $H$ with the (empirical) Fisher information matrix $F$ on the calibration dataset $D$:

$$H \approx F = \frac{1}{|D|} \sum_{d \in D} g_d g_d^T \tag{3}$$

Note that $F$ is w.r.t. a channel, differing from the diagonal Fisher information matrix in the recent works that ignores any cross-neuron interactions (Kwon et al., 2022; Kim et al., 2024).

Based on this approximation, the second order loss factor $\frac{1}{2}(c - c_0)^T(g_d g_d^T)(c - c_0)$ can be further simplified to $\frac{1}{2}(g_d^T(c - c_0))^2$, simplifying the expensive chained matrix multiplication into a single vector product. Finally, the salience can be calculated by:

$$S_c = \frac{1}{|D|} \sum_{d \in D} |g_d^T(c_q - c_0) + \frac{1}{2}(g_d^T(c_q - c_0))^2| \tag{4}$$

We do not ignore the first order information during the calculation, differing from OBD (LeCun et al., 1989) and many recent quantization works (Frantar et al., 2022; Dettmers et al., 2024; Kim et al., 2024). This is because the first order factor can be more significant in the estimation in Eq.4, as the estimated second order factor is the square of the first order factor divided by two for each sample. Considering that $g$ can be very small for the well pretrained models and the delta of the

---

**Algorithm 1** Global precision search procedure.

---

**Input:** Linear layer number $L$, weight and gradient of all linear layers ($W_i \in \mathbb{R}^{O,I}$, $G_i \in \mathbb{R}^{O,I}$ for layer $i$).
**Output:** Global channel index with large and small bit width (largebit_channels, smallbit_channels).
1: $S_{global} \leftarrow []$
2: **for** $i = 1, 2, ..., L$ **do**
3:     $W_i^{delta} \leftarrow$ quantize($W_i$) - $W_i$
4:     $S_{1st} \leftarrow$ sum($G_i \odot W_i^{delta}$, dim=1)       ▷ Per-channel dot product between $G_i$ and $W_i^{delta}$
5:     $S_{2nd} \leftarrow 0.5 * (S_{1st})^2$
6:     $S \leftarrow |S_{1st} + S_{2nd}|$             ▷ $S \in \mathbb{R}^O$, the salience of the $O$ channels
7:     **for** $channel\_id = 1, 2, .., O$ **do**     ▷ Log the salience of each output channel of this layer
8:         $S_{global}$.append(tuple($i$, $channel\_id$, $S[channel\_id]$))
9: sort($S_{global}$)                                     ▷ Sort according to the salience, descending
10: largebit_channels, smallbit_channels $\leftarrow S_{global}[: N_{largebit}]$, $S_{global}[N_{largebit} :]$

---

quantized weight is usually not large, the first order factor can be larger than the second order one. Besides, what we require is the loss itself rather than the arguments of the loss function, and thus we do not need to ignore the first order factor to simplify the arguments calculation.

Algo.1 illustrates the procedure of the global precision search. It calculates the salience of all the output channels of all linear layers and sort them in descending order globally. Given the global threshold $N_{largebit}$ as the number of large-bit precision channels, the first $N_{largebit}$ channels are intended to be quantized with 8-bit, and the other channels will be quantized with smaller bit-width (i.e., 4-bit in this paper). Any quantization methodologies (e.g., GPTQ, clip search) can be applied independently to these two disjoint parts of channels. Note that we calculate the salience of the channels in one pass rather than iterative identifying the high-salience parts in a smaller step, as we observe the single-pass method show similar results with the iterative method and saves a lot of computation overhead than the latter.

### 3.3 EFFICIENT QUANTIZATION COMPUTATION SYSTEM

**Parallel execution of sub-problems of different bit-width.** As for the execution shown in Fig.1, MixLLM puts different sub-problems onto different threads on the GPU to make them execute in parallel. Finally, the MatMul execution of the two parts write to the same target tensor with different channel indices to generate the final output. We implement this function with the fused epilogue of MatMul to scatter the output to the corresponding indices, which is basically costless.

**Two-step dequantization to make use of int8 Tensor Core.** As for the W4A8 computation, the dequantized weight and activations are $(W_q - z)s_w$ and $A_q s_a$, where $W_q$ and $z$ are uint4 datatype, $A_q$ is int8 datatype, and $s_w$ and $s_a$ are float16 datatype. Directly dequantizing the tensors into float16 datatype before the MatMul computation will prevent us using the fast 8-bit Tensor Core on the GPU. Instead, MixLLM uses a two step dequantization within each group. Specifically, MixLLM first partially dequantizes the weight into $(W_q - z)$, and then multiply it by $A_q$ with the 8-bit Tensor Core. Finally, it multiplies this MatMul result by the two scales within each group. Note that we use int8 datatype for $(W_q - z)$ that covers the data range correctly.

**Fast Int to Float conversion with partially fusing into Tensor Core instruction.** In the above two-step dequantization computation, the step 2 is the MatMul between the integer tensor $A_q(W_q - z)$ and the float tensor $s_a s_w$. It requires the integer to float conversion (I2F) before the multiply operation. The I2F instruction is expensive on the modern GPUs. Instead, we make use of the range-dependent fast I2F transformation to convert the I2F instruction into two add/sub instructions[4]. Specifically, it is based on the fact that there exists a certain range where an integer value's binary is the same as its float binary. We can add a bias to this value to make it within this range, and then subtract the bias in float (same underling binary) to restore its value in float type:

---

```
1 int tmp = src_int + bias_int;
2 int dst_float = *((float *)&tmp) - bias_fp;
```

---

[4](CUTLASS, Cited Sep. 2024) also has an implementation of fast I2F for general purpose.

```
3  // e.g., bias_int = 1262485504, bias_fp = 12582912.0f
```

We further fuse the integer subtraction into the Tensor Core mma (Matrix Multiply-Accumulate) instruction. The mma instruction computes $D = AB + D$ during the MatMul computation. We initialize the accumulator $D$ as the `bias_int` before MatMul computation of each quantization group, and will only need to subtract the `bias_float` after the MatMul. In another word, the expensive I2F is converted into a single float subtraction. The above I2F simplification brings more than 20 TOPS performance improvement for the 512/4096/4096 (M/N/K) quantized MatMul computation on an A100 GPU.

## 4 EVALUATION

### 4.1 SETUP

As for MixLLM evaluation in this paper, we use 0%, 10%, 20%, 50% and 100% percent of 8-bit based on the 4-bit quantization, respectively. Meanwhile, we use 8-bit for activation quantization. Both the weight and activation are group-wise quantized with group size 128. The 4-bit part is asymmetric quantized and the 8-bit part (including that in weight) is symmetric, which is a good trade-off between accuracy and system efficiency. Note that any other bit-width percentage configuration can be used for real scenarios to trade-off memory usage, system efficiency and accuracy in practice. We enable GPTQ (without reorder) and clip search in MixLLM for the models except for Llama 3 8B, as which shows poor performance without reorder in GPTQ. We also disable the clip search for the 70B and 72B models as the clip search tasks too long time.

**Baselines and configurations.** We compare MixLLM with the state-of-the-art (SOTA) quantization solutions of both weight-only and weight-activation methods. As for the weight only quantization, we compare MixLLM with the basic round-to-nearst (RTN) 4-bit and 5-bit quantization, and the production-level SOTA GPTQ (Frantar et al., 2022) and AWQ(Lin et al., 2024a). As for the weight-activation quantization, we compare MixLLM with the most widely used SmoothQuant (Xiao et al., 2023) and the recent SOTA QoQ (Lin et al., 2024b). The 8-bit tensors are all symmetric quantized in all baselines and MixLLM. We also compare the perplexity with SqueezeLLM(Kim et al., 2024), OminiQuant(Shao et al., 2024), AffineQuant(Ma et al., 2024), QuaRot(Ashkboos et al., 2024), Atom(Zhao et al., 2024) and SpinQuant(Liu et al., 2024) according to their reported numbers.

We make use of AutoGPTQ lib (AutoGPTQ, Cited Sep. 2024) (v0.8.0) to evaluate GPTQ, AutoAWQ lib (AutoAWQ, Cited Sep. 2024) (v0.2.6) to evaluate AWQ, and lmquant lib (MIT-Han-Lab, Cited Sep. 2024a) (commit 58a3a16) to evaluate SmoothQuant and QoQ. We enable the reorder trick for GPTQ evaluation, and use asymmetric and group size 128 for both GPTQ and AWQ. We follow the official configurations in lmquant to use 0.85/0.15 as the alpha/beta parameter for SmoothQuant, and 0.3/0.7 for QoQ. We disable the KV quantization of QoQ in our experiments to make the comparison fair.

**Models and Datasets.** We evaluate MixLLM and the baselines on a variety of widely used LLMs of different sizes, ranging from 1.5B to 72B. The models include Llama 3 8B and 70B (Meta, Cited Sep. 2024), Llama 2 7B (Touvron et al., 2023), Mistral 7B v0.3 (Jiang et al., 2023), Qwen2 1.5B, 7B and 72B (Yang et al., 2024).

We use wikitext2 dataset (Merity et al., 2017) as the calibration set for GPTQ and MixLLM. We use the default pile dataset (MIT-Han-Lab, Cited Sep. 2024b) as the calibration dataset for AWQ, SmoothQuant and QoQ, to enable their better performance. GPTQ, AWQ and MixLLM uses 128 samples with sequence length of 2048 for calibration. SmoothQuant and QoQ uses 64 samples with sequence length of 1024 for calibration (larger dataset results in OOM in our experiment).

**Metrics.** As for the algorithm accuracy, we compare the perplexity (ppl) between all the baselines on wikitext2 and C4 (Raffel et al., 2020) dataset. Meanwhile, we compare a set of popular zero shot tasks on Llama 3 8B, Mixtral 7B v0.3 and Qwen2 1.5B, including Piqa (PQ) (Tata & Patel, 2003), ARCe/ARCc (Boratko et al., 2018), BoolQ (BQ) (Clark et al., 2019), HellaSwag (HS) (Zellers et al., 2019), and WinoGrande (WG) (Sakaguchi et al., 2020).

We conduct the system experiments on NVIDIA A100 (80G) GPUs with CUDA 12.1. We use PyTorch 2.4.1 and transformers 4.42.0.

Table 1: Perplexity evaluation (↓) on wikitext2 and c4, sequence length 2048.

(a) Perplexity on wikitext2.

| baselines | | Llama 3 | | Llama 2 | Mistral | Qwen2 | | |
| | | 8B | 70B | 7B | 7B v0.3 | 1.5B | 7B | 72B |
| --- | --- | --- | --- | --- | --- | --- | --- | --- |
| float16 | | 6.14 | 2.85 | 5.47 | 5.32 | 9.54 | 7.14 | 5.22 |
| RTN | W4A16 | 6.73 | 3.72 | 5.73 | 5.51 | 10.17 | 7.46 | 5.31 |
| | W5A16 | 6.30 | 2.97 | 5.54 | 5.38 | 9.69 | 7.23 | 5.24 |
| GPTQ | W4A16 | 6.46 | - | 5.59 | 5.49 | 9.81 | 7.31 | 5.48 |
| AWQ | W4A16 | 6.55 | 3.26 | 5.60 | 5.44 | 10.09 | 7.32 | 5.28 |
| SmoothQuant | W8A8 | 6.24 | 2.97 | 5.51 | 5.34 | 9.67 | 7.26 | 5.27 |
| QoQ | W4A8 | 6.56 | 3.46 | 5.62 | 5.44 | - | - | - |
| MixLLM | W4A8 (0% 8bit) | 6.91 | 3.25 | 5.72 | 5.41 | 9.81 | 7.24 | 5.28 |
| | W4.4A8 (10% 8bit) | 6.32 | 3.04 | 5.55 | 5.36 | 9.66 | 7.18 | 5.25 |
| | W4.8A8 (20% 8bit) | 6.25 | 2.99 | 5.52 | 5.34 | 9.62 | 7.17 | 5.24 |
| | W6A8 (50% 8bit) | 6.20 | 2.90 | 5.50 | 5.33 | 9.58 | 7.15 | 5.23 |
| | W8A8 (100% 8bit) | 6.15 | 2.86 | 5.48 | 5.32 | 9.55 | 7.15 | 5.22 |

(b) Perplexity on c4.

| baselines | | Llama 3 | | Llama 2 | Mistral | Qwen2 | | |
| | | 8B | 70B | 7B | 7B v0.3 | 1.5B | 7B | 72B |
| --- | --- | --- | --- | --- | --- | --- | --- | --- |
| float16 | | 8.88 | 6.73 | 6.97 | 7.84 | 12.66 | 9.90 | 7.61 |
| RTN | W4A16 | 9.64 | 7.94 | 7.25 | 8.04 | 13.43 | 10.32 | 7.70 |
| | W5A16 | 9.08 | 7.16 | 7.06 | 7.91 | 12.85 | 10.00 | 7.64 |
| GPTQ | W4A16 | 9.47 | - | 7.15 | 8.19 | 13.25 | 10.23 | 7.69 |
| AWQ | W4A16 | 9.41 | 6.98 | 7.12 | 7.98 | 13.30 | 10.15 | 7.69 |
| SmoothQuant | W8A8 | 9.02 | 6.85 | 7.02 | 7.87 | 12.81 | 10.06 | 7.68 |
| QoQ | W4A8 | 9.41 | 7.08 | 7.13 | 7.99 | - | - | - |
| MixLLM | W4A8 (0% 8bit) | 9.59 | 7.05 | 7.18 | 7.99 | 13.23 | 10.16 | 7.70 |
| | W4.4A8 (10% 8bit) | 9.21 | 6.88 | 7.08 | 7.92 | 12.91 | 10.03 | 7.65 |
| | W4.8A8 (20% 8bit) | 9.10 | 6.84 | 7.05 | 7.89 | 12.85 | 9.99 | 7.64 |
| | W6A8 (50% 8bit) | 9.00 | 6.78 | 7.01 | 7.87 | 12.77 | 9.95 | 7.63 |
| | W8A8 (100% 8bit) | 8.89 | 6.74 | 6.98 | 7.84 | 12.68 | 9.91 | 7.62 |

## 4.2 PERPLEXITY EVALUATION

**Comprehensive comparison.** Tab.1 shows the perplexity on Wikitext2 and C4 dataset for the commonly used open source LLMs, of different baselines. GPTQ and QoQ fails to optimize some items, for which we use "-" in the table. It shows that:

- Using 4.8 bits of weights with MixLLM can outperform the 5 bits RTN quantization, even with 8-bit activation quantization enabled in MixLLM. This is mainly because MixLLM assigns the high-salience output channels with larger bit-widths than the uniform 5-bit solution.

- As for the weight-only quantization baselines, MixLLM W4.4A8 outperforms the production SOTA solutions GPTQ and AWQ, with only 10% more bit-width, and even with 8-bit activation quantization enabled in MixLLM. Meanwhile, the RTN W5A16 method also outperforms GPTQ and AWQ, which means a slightly larger bit-width can defeat the well tuned smaller bit-width easily. MixLLM W4.4A8 benefits from the larger bits on the top 10% output features with high salience.

- As for the weight-activation quantization baselines, MixLLM W4.4A8 shows a comparable accuracy with SmoothQuant with much smaller bit-width (60% of that in SmoothQuant). MixLLM W4.4A8 shows better accuracy than QoQ with only 10% larger bit-width. It shows MixLLM achieves a good balance of memory consumption and accuracy.

Table 2: Zero-shot tasks evaluation (↑) on Llama 3B, Mistral 7B v0.3 and Qwen2 1.5B.

| Models | Baselines | PQ | ARCe | ARCc | BQ | HS | WG | avg. |
|---|---|---|---|---|---|---|---|---|
| Llama 3 8B | FP16 | 80.85 | 77.78 | 53.50 | 81.31 | 79.15 | 72.61 | 74.20 |
| | GPTQ W4A16 | 80.74 | 77.74 | 51.71 | 81.07 | 78.11 | 73.64 | 73.84 |
| | AWQ W4A16 | 79.92 | 77.27 | 53.07 | 81.16 | 78.49 | 73.32 | 73.87 |
| | SmoothQuant W8A8 | 80.14 | 77.61 | 52.65 | 81.07 | 78.95 | 73.01 | 73.91 |
| | QoQ W4A8 | 80.03 | 77.86 | 51.88 | 80.12 | 78.18 | 73.64 | 73.62 |
| | MixLLM W4.8A8 | 80.20 | 79.12 | 53.07 | 79.82 | 78.69 | 73.40 | 74.05 |
| Mistral 7B v0.3 | FP16 | 82.26 | 78.24 | 52.22 | 82.11 | 80.43 | 73.80 | 74.84 |
| | GPTQ W4A16 | 81.28 | 78.03 | 51.88 | 81.35 | 79.45 | 73.01 | 74.17 |
| | AWQ W4A16 | 81.28 | 77.53 | 50.60 | 80.92 | 79.69 | 73.09 | 73.85 |
| | SmoothQuant W8A8 | 81.83 | 78.24 | 52.56 | 81.80 | 80.16 | 73.24 | 74.64 |
| | QoQ W4A8 | 82.05 | 77.86 | 51.54 | 81.50 | 79.95 | 73.88 | 74.46 |
| | MixLLM W4.8A8 | 82.05 | 77.90 | 51.71 | 82.54 | 80.05 | 73.64 | 74.65 |
| Qwen2 1.5B | FP16 | 75.41 | 60.35 | 36.09 | 72.75 | 65.41 | 65.98 | 62.67 |
| | GPTQ W4A16 | 74.43 | 59.72 | 36.18 | 71.16 | 64.54 | 64.48 | 61.75 |
| | AWQ W4A16 | 75.08 | 58.92 | 35.92 | 72.29 | 63.99 | 64.88 | 61.85 |
| | SmoothQuant W8A8 | 75.46 | 60.61 | 36.60 | 72.23 | 65.24 | 66.54 | 62.78 |
| | QoQ W4A8 | 50.82 | 25.88 | 26.91 | 37.83 | 26.85 | 51.78 | 36.68 |
| | MixLLM W4.8A8 | 75.35 | 61.78 | 36.18 | 72.42 | 64.65 | 65.90 | 62.71 |

Table 3: Effect of GPTQ and clipping (ppl ↓).

| Models | W4A16 | MixLLM w/o GPTQ&clip | MixLLM w/ GPTQ&clip |
|---|---|---|---|
| Mistral 7B v0.3 | 5.51 / 8.04 | 5.36 / 7.90 | 5.34 / 7.89 |
| Qwen2 1.5B | 10.17 / 13.43 | 9.71 / 12.89 | 9.62 / 12.85 |
| Qwen2 7B | 7.46 / 10.32 | 7.21 / 10.00 | 7.17 / 9.99 |

- Note that MixLLM W8A8 quantization (equal to the group-wise RTN quantization of both weight and activation) shows nearly lossless performance compared to the float16 baseline. This is part of the motivation that MixLLM uses group-wise quantization for the activation.

### 4.3 ZERO-SHOT TASKS EVALUATION

Tab.2 shows the accuracy of the zero-shot tasks on two popular small LLMs and a tiny model. The result shows that:

- MixLLM outperforms the weight-only quantizations. For example, for Llama 3 8B model, MixLLM shows an accuracy drop of 0.15 on average while GPTQ/AWQ show 0.36/0.33 drop.

- MixLLM shows similar, or even better, accuracy when compare with SmoothQuant. Note that the ppl metric of MixLLM is a little bit inferior to SmoothQuant, but the zero-shot metrics of MixLLM are superior than SmoothQuant on Llama 3 8B and Mistral 7B v0.3. This can partially come from the group-wise quantized activation of MixLLM. As for the tiny 1.5B model, both MixLLM and SmoothQuant show comparable accuracy with the float16 baseline.

- MixLLM shows better accuracy than QoQ on average, and better on most of the single tasks.

**Ablation study on the effect of GPTQ and clipping.** Tab.3 compares the perplexity of enabling GPTQ&clipping and disabling GPTQ&clipping. We use the Mistral and Qwen2 model for the comparison because they usually do not effect by the reorder trick of GPTQ, while Llama 2/3 models are sensitive to the reorder trick. Note we do not enable the reorder for the GPTQ in MixLLM. It shows that even though the GPTQ and clipping contributes a little to the final accuracy, the main accuracy gain comes from the mix-precision than the pure 4-bit.

**Ablation study of non-diagonal Fisher Information Matrix (FIM) and not ignoring first-order derivative.** We have two small optimization decisions in Sec.3.2: using non-diagnal FIM (refer to `opt-1` in this section), and ignoring first-order derivative (refer to `opt-2` in this section). We use Llama 3 8B model to validate this two optimizations of precision search. We disable GPTQ and

Table 4: Ablation study of using non-diagonal FIM (opt-1) and not ignoring first-order derivative (opt-2) in global precision search.

| Dataset | opt-1 off, opt-2 off | opt-1 off, opt-2 on | opt-1 on, opt-2 off | opt-1 on, opt-2 on |
|---|---|---|---|---|
| wikitext2 | 6.400 | 6.416 | 6.401 | 6.398 |
| c4 | 9.211 | 9.240 | 9.212 | 9.210 |

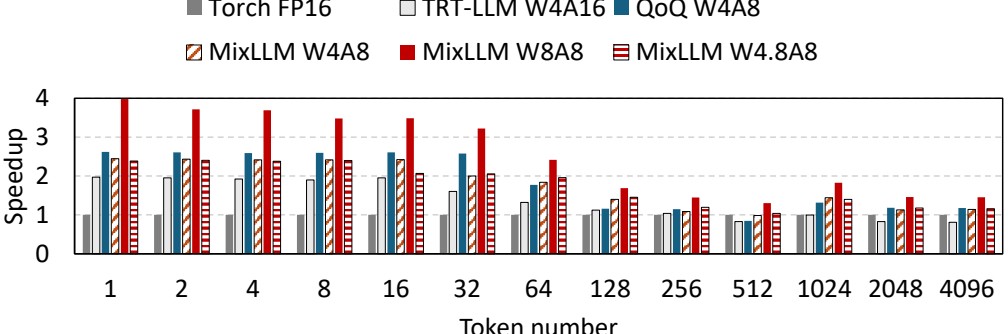

Figure 2: The speedup of a single linear layer over torch float16 baseline on the A100 GPU.

clip, and enable/disable each of the two optimizations to measure the perplexity, shown in Tab.4. It shows interesting results that, when one optimization is turned off, turning on the other optimization will hurt the accuracy. However, when one optimization is on, turning on the other one will increase the accuracy. Turning both optimizations on (as in MixLLM) will get the best accuracy.

### 4.4 SYSTEM PERFORMANCE

We have evaluated MixLLM for the single linear layer of token number ranging from 1 to 4096 with hidden size 4096, and compared it with the SOTA W4A16 (TRT-LLM) and QoQ (Lin et al., 2024b), shown in Fig.2. It also shows MixLLM kernel performance of different percent of 8-bits (W4A8 0% 8-bit, W4.8A8 20% 8-bit, and W8A8 100% 8-bit). It shows that:

- MixLLM outperforms the float16 counterpart for all token numbers, achieving $1.78\times$, $2.55\times$, and $1.77\times$ averaged speedup with MixLLM W4A8, W8A8, and W4.8A8 respectively.

- MixLLM outperforms the SOTA W4A16 solution, achieving $1.28\times$, $1.78\times$, and $1.29\times$ averaged speedup with MixLLM W4A8, W8A8, and W4.8A8 respectively.

- MixLLM achieves similar performance with QoQ with similar bit-width, achieving $0.99\times$, $1.37\times$, and $1.00\times$ averaged speedup with MixLLM W4A8, W8A8, and W4.8A8 respectively. Note that MixLLM has better accuracy than QoQ (Tab.1, Tab.2).

## 5 SUMMARY

We have presented MixLLM, achieving high accuracy with low memory consumption and high system efficiency with the rarely explored optimization space of mixed-precision quantization between output features. MixLLM identifies the salience of each output feature according to the loss distance estimation w.r.t. the global model loss rather than local layer loss. By assigning larger bit-width to the features need it most, MixLLM achieves the superior accuracy to SOTA with low memory consumption. The sub-problems of different bit-widths are disjoint and can run in parallel efficiently on the GPU. We have identified the sweet spot of the quantizaiton configuration that is friendly to both accuracy and system efficiency. To address the challenge of system efficiency, we design the two-step dequantization to enable using int8 Tensor Core computation and the fast integer-float conversion to reduce the dequantization overhead. Experiment results show that MixLLM achieves superior accuracy to SOTA with low memory cost and high system efficiency.

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

# A  APPENDIX

## A.1  COMPARISON WITH MORE RELATED WORKS

Table 5: PPL (wikitext2) comparison with more works, '-G' means GPTQ enabled in the baselines.

| Llama | FP16 | SqueezeLLM W4A16 0.45% | OmniQuant W4A16/A4 | AffineQuant W4A16/A4 | QuaRot-G W4A16/A4 | Atom W4A4 | SpinQuant-G W4A16/A4 | MixLLM W4.8A8 |
|---|---|---|---|---|---|---|---|---|
| 2-7B | 5.47 | 5.57 | 5.58 / 14.26 | 5.58 / 12.69 | 5.60 / 6.10 | 6.03 | 5.6 / 5.9 | 5.52 |
| 3-8B | 6.14 | - | - | - | - | - | 6.4 / 7.1 | 6.25 |

We compare MixLLM with more recent quantization works according to the reported numbers in their papers (Tab.5), showing that MixLLM achieves superior accuracy to a broad range of related works with similar memory consumption.

Table 6: The overhead of global precision search in MixLLM.

| Models | Llama 3 | | Llama 2 | Mistral | Qwen2 | | |
| | 8B | 70B | 7B | 7B v0.3 | 1.5B | 7B | 72B |
|---|---|---|---|---|---|---|---|
| Time (min) | 7 | 55 | 7 | 7 | 2 | 7 | 57 |

### A.2 One-pass vs. Progressive Search

As described in Sec.3.2, MixLLM searches the high-salience features within a single pass. We have tried the progressive procedure on Llama 2 7B and Mistral 7B models, which identifies smaller portions of the high-salience features iteratively. Results show that the accuracy is the same to the one-pass method to two decimal places. However, the progressive method shows much higher search time due to the repeated procedure. The one-pass method takes 7 minutes for each of the two models to search 10% high-salience features, while the progressive method that searches 2% high-salience iteratively takes 30 minutes to find top 10% features.

### A.3 Overhead of Global Precision Search

Tab.6 shows the global precision search overhead described in Sec.3.2. As noted in Sec.4.1, the calibration dataset has 128 samples with sequence length of 2048. We use a single A100 GPU for the 1.5B, 7B and 8B models, and 4 A100 GPUs for the 70B and 72B models to perform the search. We make use of `device_map` in huggingface for multi-GPU execution, which is sequential execution of layers. The 7B models require about 7 minutes and the 70B models require less than 60 minutes to complete the search. Considering that the quantization only needs to be preformed once, the searching algorithm is practical for the real workloads.

### A.4 Performance challenge of the float16 outlier separation

Outlier separation with half precision works to improve the accuracy while using small bit-width for the non-sensitive weights (Kim et al., 2024; Dettmers et al., 2024), by separating the outliers into an extra sparse tensor in float16. However, it is hard to achieve the peak performance due to the inefficiency of the sparse computation on the GPU, especially when the batch size is large and the linear layer becomes compute-bounded. (As discusseded in Flash-LLM (Xia et al., 2023), the hardware utilization can be lower than 10% for the sparse MatMul, while its dense counterpart can usually achieve more than 60%.) This is because the unstructured tensor computation cannot make use the fast Tensor Core easily, but has to use the SIMT Core in float16 for computation and float32 for accumulation[5]. Note the peak performance of int8 Tensor Core is $8\times$ higher than that of float16 SIMT Core on A100 GPU (NVIDIA, Cited Sep. 2024), and $32\times$ higher than float32 SIMT Core. Moreover, sparse computing makes it more difficult to fully utilize the hardware due to the non-continuous memory pattern and the extra index computation.

---

[5]Flash-LLM (Xia et al., 2023) optimizes the unstructured sparse MatMul, but can only speedup the small-batched scenarios.

