# OpenReview forum: "MixLLM: Mixed-precision LLM Quantization with Algorithm-system Co-design"
_ICLR.cc/2025/Conference — ICLR 2025 Conference Withdrawn Submission_

### Official Review · Reviewer_VuL3 · 2024-10-31

**Soundness:** 3
**Presentation:** 2
**Contribution:** 3
**Rating:** 6
**Confidence:** 3

**Summary:**

This paper presents a method called MixLLM, aimed at achieving high accuracy and system efficiency through mixed-precision quantization. MixLLM introduces to identify the salience of output features based on loss distance estimates, focusing on global model loss rather than local layer loss. By assigning larger bit-widths to the most critical features, MixLLM achieve superior while maintaining low memory consumption. This paper presents a two-step dequantization process to reduce overhead and efficiently utilize int8 Tensor Core for computation. Experimental results demonstrate the effectiveness of MixLLM across various tasks compared to existing techniques.

**Strengths:**

1. The introduction of MixLLM explores a novel optimization space in mixed-precision quantization through global salience identification.
2. The paper clearly articulates how the two-step dequantization improves computational efficiency, particularly through leveraging Tensor Core capabilities, demonstrating both practicality and usability.
3. The authors conduct extensive experiments across multiple tasks and models, providing compelling evidence for the advantages of MixLLM in terms of accuracy and efficiency.

**Weaknesses:**

1. Although the study evaluates several models and datasets, the comparisons with some state-of-the-art methods are not sufficiently comprehensive, potentially limiting the generalizability of the results.
2. The provided Algorithm 1 is not standardized. It should clearly specify the input and output, adhere to a standard for statement format, and include step numbers for each procedure.
3. Provide detailed experimental setups, hyperparameters, and implementation specifics to Enhance Reproducibility.

**Questions:**

It would be better to expand the discussion on theoretical analysis behind MixLLM and the limitations of MixLLM.

---

> ### Author Response · Authors · 2024-11-26
>
> Thank you very much for your thoughtful review and constructive suggestions. We have tried to address your questions carefully and revised the manuscript thoroughly. We hope the reviewer will consider raising your score in light of our response.
>
> ## W1&W3&Q1: more comparison of both algorithm and system, and more discussion
> - More accuracy comparison, two more larger models and different ratios of 8-bits.
>   - We have added the accuracy comparison of LLaMA 3 70B and Qwen2 72B in Tab.1 to show its efficacy on larger models. We have also added the accuracy of different ratio of 8-bits, including 0%, 10%, 20%, 50% and 100% 8-bits, shown in L391-395 and L407-411 in the revised paper. MixLLM still shows higher accuracy than the SOTA W4A16 and W4A8 solutions with only 10% of 8-bits (W4.4A8).
> - More system performance evaluation
>   - We have revised Sec.4.4 to add system performance data of linear layer execution (Fig.2 and L516-527), with a wide range of token numbers (from 1 to 4096). We have the following summary of the single linear layer performance:
>     -  MixLLM outperforms the float16 counterpart for all token numbers, achieving 1.78$\times$, 2.55$\times$, and 1.77$\times$ averaged speedup with MixLLM W4A8, W8A8, and W4.8A8 respectively.
>     - MixLLM outperforms the SOTA W4A16 solution, achieving 1.28$\times$, 1.78$\times$, and 1.29$\times$ averaged speedup with MixLLM W4A8, W8A8, and W4.8A8 respectively.
>     - MixLLM achieves similar performance with QoQ with similar bit-width, achieving 0.99$\times$, 1.37$\times$, and 1.00$\times$ averaged speedup with MixLLM W4A8, W8A8, and W4.8A8 respectively. Note that MixLLM has better accuracy than QoQ.
> - More ablation study and analysis
>   - _Ablation study of (1) use non-diagonal Fisher information matrix and (2) not ignoring first-order information._ We have added the ablation study in L483-L512 and Tab.4. It shows interesting results that, when one optimization is turned off, turning on the other optimization will hurt the accuracy. However, when one optimization is on, turning on the other one will increase the accuracy. Turning both optimizations on (as in MixLLM) will get the best accuracy.
>   - _One-pass vs. progressive search._ We have added the discussion in Appendix A.2, L763-771.We have tried the progressive procedure on Llama 2 7B and Mistral 7B models, which identifies smaller portions of the high-salience features iteratively. Results show that the accuracy is the same to the one-pass method to two decimal places. However, the progressive method shows much higher search time due to the repeated procedure. The one-pass method takes 7 minutes for each of the two models to search 10% high-salience features, while the progressive method that searches 2% high-salience iteratively takes 30 minutes to find top 10% features.
>   - _Quantization overhead._ We have evaluated the overhead of the global precision search and added the new section in Appendix A.3. We use a single A100 GPU for the 1.5B, 7B and 8B models, and 4 A100 GPUs for the 70B and 72B models to perform the search. We make use of device_map in huggingface for multi-GPU execution, which is sequential execution of layers on different devices. The 7B models require about 7 minutes and the 70B models require less than 60 minutes to complete the search. Considering that the quantization only needs to be performed once, the searching algorithm is practical for the real workloads.
> - We have revised the setup section and the ablation sections to provide detailed experimental setups and hyperparameters.
>
> ## W2: Algorithm 1 is not standardized
> - Thanks for the suggestion. We have revised Algorithm 1, specified the input and output and rewritten to a standard for statement format.

---

### Official Review · Reviewer_zKpt · 2024-11-01

**Soundness:** 2
**Presentation:** 2
**Contribution:** 3
**Rating:** 5
**Confidence:** 4

**Summary:**

This paper address the performance degradation that often occurs when LLMs are quantized to low-bit representations. The authors propose a mixed-precision quantization method for model weights, specifically applying both int4 and int8 quantization to the output channels of the weights. The authors introduce an approach based on global loss estimation and second-order Taylor expansion. This method pinpoints the output channels that are most sensitive to quantization errors, allowing the allocation of higher precision to those critical channels. In addition, the authors design a fast i2f conversion technique to enhances the model's inference speed. Extensive experiments, the paper demonstrates that using 20% int8 quantization  can achieve comparable performance  with full-precision models.

**Strengths:**

1. The paper introduces a novel approach to pinpoint output channels that are difficult to quantize by utilizing global loss estimation and second-order Taylor expansion. This method allows for effective mixed-precision quantization of weights, specifically applying int4 and int8 to the output channels, which significantly reduces quantization loss without adding extra computational overhead.

2.  The design of a fast integer-to-float (i2f) conversion method is a noteworthy innovation. By leveraging GPU int8 computational capabilities and minimizing the latency typically associated with i2f conversions, the authors enhance computational efficiency, which is critical for high-performance applications.

3. The paper is well-written, with clear descriptions and thorough explanations of the methods used. The authors provide comprehensive experiments that demonstrate their approach's effectiveness, showing that using int8 quantization on just 20% of the channels can achieve state-of-the-art results.

**Weaknesses:**

1. Limited Model Size in Experiments: The models tested in the paper are relatively small. The absence of experiments on larger models, such as those with 30B or 70B parameters, raises questions about the scalability of the proposed method. Evaluating larger models would strengthen the paper by demonstrating the approach's effectiveness across a broader range of model sizes.

2. Potential Efficiency Concerns: The quantization process requires performing global loss calculations and obtaining gradients, which may be computationally intensive. This could pose efficiency challenges, especially when dealing with very large models, and might offset some of the benefits gained from the quantization technique.

3. Hardware Dependency: The implementation relies on specific w8a8 computational kernels to realize the performance gains. As a result, the acceleration may be significant only on certain GPUs that support these optimizations. This hardware dependency could limit the method's general applicability and usefulness across different computing environments that do not have the requisite hardware support.

**Questions:**

1. Experiments on Larger Models: The models tested in your work are relatively small in scale (up to 8B LLM). Conduct experiments on larger models, such as those with 30B or 70B is expected to demonstrate the scalability and effectiveness.

2. Quantization Time Efficiency: This method involves calculating global loss and obtaining gradients to identify the output channels requiring higher bit-widths. Could you provide more details on the time efficiency of this quantization process? Specifically, how does the computational overhead compare to other quantization techniques, and does it significantly impact the overall efficiency when scaling to larger models?

3. Latency Testing on Different GPUs: Given that your optimizations are designed to leverage GPU capabilities, have you conducted latency tests on different GPU architectures? Providing performance benchmarks across various GPUs would offer valuable support into the general applicability and potential limitations of your method in diverse hardware environments.

4. Release of CUDA Kernel Code: To facilitate community validation and further research, are you planning to release the CUDA kernel code for your custom w8a8 computation? Sharing the code would enable others to replicate your results, integrate your optimizations into their own work, and contribute to advancements in this area.

---

> ### Author Response · Authors · 2024-11-26
>
> Thank you very much for your thoughtful review and constructive suggestions. We have tried to address your questions carefully and revised the manuscript thoroughly. We hope the reviewer will consider raising your score in light of our response.
>
> ## W1&Q1: experiments on larger models.
> - We have added the accuracy comparison of LLaMA 3 70B and Qwen2 72B in Tab.1. We have also added the accuracy of different ratio of 8-bits, including 0%, 10%, 20%, 50% and 100% 8-bits, shown in L391-395 and L407-411 in the revised paper. MixLLM still shows higher accuracy than the SOTA W4A16 and W4A8 solutions with only 10% of 8-bits (W4.4A8).
>
> ## W2&Q2: quantization overhead analysis.
> - We have evaluated the overhead of the global precision search and added the new section in Appendix A.3. We use a single A100 GPU for the 1.5B, 7B and 8B models, and 4 A100 GPUs for the 70B and 72B models to perform the search. We make use of _device\_map_ in huggingface for multi-GPU execution, which is sequential execution of layers on different devices. The 7B models require about 7 minutes and the 70B models require less than 60 minutes to complete the search. Considering that the quantization only needs to be performed once, the searching algorithm is practical for the real workloads.
>
> ## W3&Q3: latency testing
> - We have revised Sec.4.4 to add system performance data of linear layer execution (Fig.2 and L516-527), with a wide range of token numbers (from 1 to 4096). We have the following summary of the single linear layer performance:
>   - MixLLM outperforms the float16 counterpart for all token numbers, achieving 1.78$\times$, 2.55$\times$, and 1.77$\times$ averaged speedup with MixLLM W4A8, W8A8, and W4.8A8 respectively.
>   - MixLLM outperforms the SOTA W4A16 solution, achieving 1.28$\times$, 1.78$\times$, and 1.29$\times$ averaged speedup with MixLLM W4A8, W8A8, and W4.8A8 respectively.
>   - MixLLM achieves similar performance with QoQ with similar bit-width, achieving 0.99$\times$, 1.37$\times$, and 1.00$\times$ averaged speedup with MixLLM W4A8, W8A8, and W4.8A8 respectively. Note that MixLLM has better accuracy than QoQ.
> - The system optimization of MixLLM can be applied to any architectures supporting int8 computation, which is supported on nearly all the modern GPUs. We implement MixLLM kernels based on CUTLASS API, which supports different NVIDIA GPUs well.
>
> ## Q4: code release
> - Yes, we will opensource the algorithm and CUDA kernels in the future. We believe it will benefit a broad range of LLM applications. Thanks.

---

> > ### Comment · Reviewer_zKpt · 2024-12-03
> >
> > Thank you for the author's detailed response. I have carefully read your replies, and I appreciate the efforts you have made to address my concerns. Most of my questions have been resolved.
> >
> > However, I still believe that the comparison methods seem somewhat outdated. It would be beneficial to include comparisons with more advanced method, such as Quarot[1], to provide a richer set of experiments that demonstrate the advantages of your proposed method. Therefore, I will maintain my original score.
> >
> > Additionally, I suggest that you clearly indicate the changes made in the manuscript to help reviewers easily identify the main modifications. This would greatly improve the clarity of your revisions.
> >
> > [1] QuaRot: Outlier-Free 4-Bit Inference in Rotated LLMs, NeurIPS 2024

---

> ### Author Response · Authors · 2024-12-03
>
> Thank you for your recognition that our revision solves most of your concerns!
>
> We do have the comparison to a wide range of related works, including Quarot and 5 more other works (SqueezeLLM, OminiQuant, AffineQuant, Atom, and SpinQuant). These comparisons are shown in Tab.5 in the Appendix section. The results show that MixLLM has superior accuracy than all these related works on Llama 2 7B. Note that the code of most of these works does not support Llama 3 and Qwen models natively so that is hard to put in Tab.1. For example, the official github repo of Quarot claims that they only support Llama 2 models: https://github.com/spcl/QuaRot/tree/main/fake_quant .
>
> Besides, QoQ[1] is a more recent work that has shown better accuracy and system efficiency than Quarot and Atom, and thus we mainly compare our work to QoQ in Tab.1 and Tab.2.
>
> We will highlight the above information in the next version. We hope it can further address your concerns.
>
> Thanks again for your constructive comments that makes the paper better!
>
> [1] QServe: W4A8KV4 Quantization and System Co-design for Efficient LLM Serving. Yujun Lin, Haotian Tang, Shang Yang, Zhekai Zhang, Guangxuan Xiao, Chuang Gan, Song Han.

---

### Official Review · Reviewer_LhDa · 2024-11-03

**Soundness:** 2
**Presentation:** 2
**Contribution:** 2
**Rating:** 6
**Confidence:** 4

**Summary:**

This paper proposes a new optimization approach for mixed precision quantization of output features, recognizing that different features have varying importance in the model. By allocating larger bit widths to the most critical features, MixLLM enhances accuracy while minimizing memory usage. To improve system efficiency, this paper also implements a two-step dequantization process, allowing the use of int8 Tensor Core computations and fast integer-float conversion to reduce dequantization overhead. The results demonstrate that MixLLM outperforms QoQ while maintaining similar memory consumption and system efficiency.

**Strengths:**

This paper achieves better results on various LLMs, including Llama 3 8B, Llama 2 7B, Mistral 7B v0.3, Qwen 2 1.5B, and Qwen 2 7B. It also demonstrates improved performance across various benchmarks compared to QoQ, while maintaining similar memory consumption.

**Weaknesses:**

1. In Table 1, QoQ fails to optimize the Qwen 2 1.5B and Qwen 2 7B models, leading to an unfair comparison.
2. There is a lack of comparison of MixLLM's inference time with other methods.
3. Figure 1 fails to clearly illustrate the complete process of MixLLM, and this paper includes only one figure.

**Questions:**

1.	Could you show the inference time of MixLLM in comparison to other methods under the same experimental setup?
2.	Could you provide a comparison of the performance and inference time of this method on larger models like LLaMA3 70B and Qwen2 72B against other methods?
3.	Could you provide the ablation experiments on performance and inference time for the 'two-step dequantization' proposed in this paper?

---

> ### Author Response · Authors · 2024-11-26
>
> Thank you very much for your thoughtful review and constructive suggestions. We have tried to address your questions carefully and revised the manuscript thoroughly. We hope the reviewer will consider raising your score in light of our response.
>
> ## W1: about the failed baseline
> - We believe the failure of QoQ on Qwen2 models is due to some minor engineer problems. In our revised version, we have removed the exact result of the failed cases, but fill "-" instead (refer to the revised Tab.1.).
>
> ## W2&Q1&Q3: MixLLM system performance
> - We have revised Sec.4.4 to add system performance data of linear layer execution (Fig.2 and L516-527), with a wide range of token numbers (from 1 to 4096). We have the following summary of the single linear layer performance:
>   - MixLLM outperforms the float16 counterpart for all token numbers, achieving 1.78$\times$, 2.55$\times$, and 1.77$\times$ averaged speedup with MixLLM W4A8, W8A8, and W4.8A8 respectively.
>   - MixLLM outperforms the SOTA W4A16 solution, achieving 1.28$\times$, 1.78$\times$, and 1.29$\times$ averaged speedup with MixLLM W4A8, W8A8, and W4.8A8 respectively.
>   - MixLLM achieves similar performance with QoQ with similar bit-width, achieving 0.99$\times$, 1.37$\times$, and 1.00$\times$ averaged speedup with MixLLM W4A8, W8A8, and W4.8A8 respectively. Note that MixLLM has better accuracy than QoQ.
>
> ## W3: illustration of the procedure
> - We have revised Algo.1 to make the precision search procedure clearer (L270-284). It calculates the salience of each output features (channel) and assigns the large bit-width to the high-salience features. After identifying the bit-width of each feature, the quantization procedure is similar to the existing works, with the unique consideration in Sec.3.3. Fig.1 shows the high-level concept of the computation, that the 4-bit and 8-bit sub-MatMul are executed independently.
>
> ## Q2: comparison with LLaMA 3 70B and Qwen2 72B.
> - We have added the accuracy comparison of LLaMA 3 70B and Qwen2 72B in Tab.1. We have also added the accuracy of different ratio of 8-bits, including 0%, 10%, 20%, 50% and 100% 8-bits, shown in L391-395 and L407-411 in the revised paper. MixLLM still shows higher accuracy than the SOTA W4A16 and W4A8 solutions with only 10% of 8-bits (W4.4A8).

---

### Official Review · Reviewer_MZ5X · 2024-11-05

**Soundness:** 2
**Presentation:** 4
**Contribution:** 3
**Rating:** 6
**Confidence:** 4

**Summary:**

This paper presents (1) a survey of existing post-training-quantization (PTQ) approaches; (2) a new approach for mixed precision based on global salience.

The proposed method defines a global salience measure of each output channel, similar to that of Kwon et al., 2022 and Kim et al., 2024 but with modifications. Different from previous works that use such salience information to perform sparse-and-dense decomposition, this paper packs output channels into two bit-rate buckets (8-bit symmetric and 4-bit asymmetric). The mixed-precision matmul is then carried out with a fused kernel which scatters output (of matmuls from different bitrate buckets) back to the corresponding indices.

**Strengths:**

* The paper does a good job summarizing and providing useful critiques to different PTQ approaches in Section 2, including weight-only quantization, weight-activation quantization, outlier separation and mixed-precision approaches. Many of the assessment maybe subjective but accurate according to the reviewer's own experience.

* This proposed method which uses  salience information to perform packing of output channels into different bit-rate bucket is interesting and novel to the knowledge of the reviewer. This process is intuitive, and requires just a simple sorting instead of using iterative procedure, which as the authors pointed out, "saves a lot of computation" (Ln269). The results looks good on 7B-sized models.

**Weaknesses:**

**The "co-design" analysis**

While the paper claims "algorithm-system co-design", it doesn't contain much principled quantitative analysis. For a good "co-design" system paper, the reviewer would expect the author to use some kind of performance models, such as the roofline model to justify why the chosen design is a "sweet spot". Or at least use performance counters such as cache hit-rate, communication latency, memory bandwidth to substantiate the choice. In the current state, the paper relies too heavily on empirical observations and many claims are not very well explained. For example,
* Ln46: "Besides, the weight-only method can lead to system performance drop for large-batched workloads." \
In what model and what's the batch size when this happens?

* Ln75: "MatMul execution tends to be bound more on the larger weight tensor rather than the smaller activation tensor, which weakens the need to push the activation smaller" \
"Weakens the need" is a quite vague expression and it will be nice to draw your conclusion in a data-driven manner.

* Ln695: "Hard to achieve the peak performance due to the inefficiency of the sparse tensor computation on the GPU"\
What is the utilization of the proposed method and how bad is the utilization on approaches that separate outliers in contrast?

**Reproducibility**

The paper currently does not show throughput / latency numbers (or did I miss it?) nor was the inference kernel implementation provided. Because MixLLM uses different bit-rate (W4.8) and custom inference kernel (packing and scattering), it is important to carefully benchmark the latency and throughput against other methods. Currently the paper only says "this function with the fused epilogue of MatMul to scatter the output to the corresponding indices, which is basically costless" (Ln288) and "MixLLM is on-pair for the system efficiency
when compared to the state-of-the-art weight-activation quantization" (Ln 466)

While Github implementation is not generally required, the reviewer notes that most quantization papers do have OSS implementation (GPTQ, AWQ, LLM.int8, SmoothQuant, QoQ). Since the actual system performance of the scattering kernel is critical to the usefulness of this paper, it would be difficult to verify the results and seriously limit the usefulness of the paper without an implementation of the fused kernel.

**Questions:**

1. In Eq 4, two design decisions that differentiate the proposed salience measure from previous work is to (1) use non-diagonal Fisher information matrix and (2) not ignoring first-order information. Can you provide ablation results show how important they are to the final results?

2. MixLLM is evaluated with one fixed setting of 20% 8-bit + 80% 4-bit. It will be nice to see how the accuracy / latency / throughput change when this ratio is changed.

3. The author makes the argument that the proposed one-pass method is faster / competitive to iterative methods. It will be nice to show how much speed-up / accuracy difference does this make.

---

> ### Author Response · Authors · 2024-11-26
>
> Thank you very much for your thoughtful review and constructive suggestions. We have tried to address your questions carefully and revised the manuscript thoroughly. We hope the reviewer will consider raising your score in light of our response.
>
> ## W1: quantitative analysis and explain the claims
> - **"Besides, the weight-only method can lead to system performance drop for large-batched workloads"**. Revised in L047-048 in the paper: the SOTA W4A16 kernel only achieves 83\% performance of its float16 counterpart at batch size 512 with hidden size 4096, shown in Fig.2.
> - **"MatMul execution tends to be bound more on the larger weight tensor rather than the smaller activation tensor, which weakens the need to push the activation smaller"**. Revised in L079 and L218-223. We calculated the compute-intensity and indicate that reducing activation from 8-bit to 4-bit does not increase the compute-intensity much: Given token number $M$ and input and output features $K$ and $N$, the compute intensity $I = \frac{2MNK}{MKB_{act} + KNB{weight}}$.
> $B_{act}$ and $B_{weight}$ are the bytes per element of activation and weight.
> Given $M = 512$ and $N = K = 4096$, reducing $B_{weight}$ from 8 to 4 will results in an 80\% increasement of $I$, while reducing $B_{act}$ from 8 to 4 will only achieve 5.88\% increasement.
> - **"Hard to achieve the peak performance due to the inefficiency of the sparse tensor computation on the GPU"**. We revised L787-789. As discussed in Flash-LLM [1], the hardware utilization can be lower than 10\% for the sparse MatMul, while its dense counterpart can usually achieve more than 60\%.
>
> ## W2: system performance and opensource
> - We have revised Sec.4.4 to add system performance data (Fig.2 and L516-527). We have the following summary of the single linear layer performance:
>   - MixLLM outperforms the float16 counterpart for all token numbers, achieving 1.78$\times$, 2.55$\times$, and 1.77$\times$ averaged speedup with MixLLM  W4A8, W8A8, and W4.8A8 respectively.
>   - MixLLM  outperforms the SOTA W4A16 solution, achieving 1.28$\times$, 1.78$\times$, and 1.29$\times$ averaged speedup with MixLLM  W4A8, W8A8, and W4.8A8 respectively.
>   - MixLLM  achieves similar performance with QoQ with similar bit-width, achieving 0.99$\times$, 1.37$\times$, and 1.00$\times$ averaged speedup with MixLLM  W4A8, W8A8, and W4.8A8 respectively. Note that MixLLM  has better accuracy than QoQ.
> - Yes, we will opensource the algorithm and CUDA kernels in the future. Thanks.
>
> ## Q1: ablation study of (1) use non-diagonal Fisher information matrix and (2) not ignoring first-order information
> - We have added the ablation study in L483-L512 and Tab.4. It shows interesting results that, when one optimization is turned off, turning on the other optimization will hurt the accuracy. However, when one optimization is on, turning on the other one will increase the accuracy. Turning both optimizations on (as in MixLLM) will get the best accuracy.
>
> ## Q2: evaluate different ratio of mixed-precision
> - Accuracy: We have added the ppl of different ratio of 8-bit, including 0%, 10%, 20%, 50% and 100% 8-bits, shown in L391-395 and L407-411 in the revised paper. We also revised the discussion in Sec.4, that only 10% of 8-bit can lead to better result than the SOTA W4A16 and W4A8 solutions.
> - System performance: the revised Sec.4.4 and newly added Fig.2 shows the linear kernel performance of 0%, 20% and 100% 8-bits in MixLLM, which shows the efficacy of MixLLM system.
>
> ## Q3: one-pass vs. progressive search
> - We have added the discussion in Appendix A.2, L763-771.We have tried the progressive procedure on Llama 2 7B and Mistral 7B models, which identifies smaller portions of the high-salience features iteratively. Results show that the accuracy is the same to the one-pass method to two decimal places. However, the progressive method shows much higher search time due to the repeated procedure.
> The one-pass method takes 7 minutes for each of the two models to search 10\% high-salience features, while the progressive method that searches 2\% high-salience iteratively takes 30 minutes to find top 10\% features.
>
>
> [1] Flash-LLM: Enabling Cost-Effective and Highly-Efficient Large Generative Model Inference with Unstructured Sparsity. Haojun Xia, Zhen Zheng, Yuchao Li, Donglin Zhuang, Zhongzhu Zhou, Xiafei Qiu, Yong Li, Wei Lin, Shuaiwen Leon Song. VLDB 2024.

---

> ### Comment · Reviewer_MZ5X · 2024-12-02
>
> The author's rebuttal addresses most of the issues I raised (ratio, latency, ablation). Thanks for the thorough change and I am revising my score to reflect the change.
>
> Some minor comments after the revision.
> * Ablation test actually shows the improvements from Opt-1 and Opt-2 is quite small.
> * W4.8A8 has the same speed as W4A8 and slower than W8A8 under small batch sizes, even though W4.8 has less memory bandwidth requirement. This indicates that the main benefits will be of smaller model size.

---

> > ### Author Response · Authors · 2024-12-03
> >
> > Thank you very much for your constructive comments and support! Your comments are very valuable in improving the paper. We appreciate your time and effort in reviewing our paper.

---

### Note · Authors · 2024-12-20

**Comment:**

Due to incomplete author information, the manuscript had to be withdrawn. We thank AC and all the reviewers for their efforts, thoughtful reviews, and constructive suggestions.

**Withdrawal Confirmation:**

I have read and agree with the venue's withdrawal policy on behalf of myself and my co-authors.